# Strongly interacting light dark matter

**Sebastian Bruggisser[1★], Francesco Riva[2†] and Alfredo Urbano[2♭]**

**1** DESY, Notkestrasse 85, D-22607 Hamburg, Germany
**2** CERN, Theoretical Physics Department, Geneva, Switzerland

★ sebastian.bruggisser@desy.de
† francesco.riva@cern.ch
♭ alfredo.leonardo.urbano@cern.ch

## Abstract

**We discuss a class of Dark Matter (DM) models that, although inherently strongly coupled, appear weakly coupled at small-energy and fulfill the WIMP miracle, generating a sizable relic abundance through the standard freeze-out mechanism. Such models are based on approximate global symmetries that forbid relevant interactions; fundamental principles, like unitarity, restrict these symmetries to a small class, in such a way that the leading interactions between DM and the Standard Model are captured by effective operators up to dimension-8. The underlying strong coupling implies that these interactions become much larger at high-energy and represent an interesting novel target for LHC missing-energy searches.**

# 1 Motivation

Studies of processes with missing energy at the LHC constitute an important part of the Dark Matter (DM) research program, that aims at unravelling possible non-gravitational interactions

between the Standard Model (SM) and the dark sector. Information from the LHC would be particularly useful for light DM, $m_{\rm DM} \lesssim 10$ GeV, below the threshold for direct detection experiments. In this case, the *WIMP miracle* seems to provide a convincing hint that light DM originates from *weakly coupled* dynamics. Indeed, parameterizing the thermally-averaged annihilation cross section as

$$\langle \sigma v_{\rm rel} \rangle \sim \frac{\alpha_{\rm DM}^2}{m_{\rm DM}^2} \,, \tag{1}$$

with $m_{\rm DM}, \alpha_{\rm DM}$ the DM mass and coupling to the Standard Model (SM) fields, we find for the relic density

$$\Omega_{\rm DM} h^2 \approx \frac{10^{-26}\,{\rm cm}^3/{\rm s}}{\langle \sigma v_{\rm rel} \rangle} \approx 0.1 \left( \frac{0.1}{\alpha_{\rm DM}} \right)^2 \left( \frac{m_{\rm DM}}{10\,{\rm GeV}} \right)^2 .$$

A weak coupling $\alpha_{\rm DM} \ll 1$ reproduces the observed value $\Omega_{\rm DM} h^2 \approx 0.1$ [1].[1]

In this letter we want to explore how solid this indication is and study the viability of light DM associated with a new *strong*, yet perturbative, coupling which we call $g_* \lesssim 4\pi$. The core aspect of our analysis is *approximate symmetries*, which forbid relevant (renormalizable) SM-DM interactions, but allow irrelevant (non-renormalizable) interactions of dimension $D$. Referring to $M$ as the *physical* scale suppressing the latter, the amplitude for $2 \to 2$ annihilation, would scale as

$$\alpha_{\rm DM} \sim \frac{g_*^2}{4\pi} \left( \frac{E}{M} \right)^{D-4} , \tag{2}$$

where $E$ denotes the collision energy. At low energies $E \ll M$, such as those relevant at freeze-out, the interaction of Eq. (2) appears weak, despite their strongly coupled nature at high-energy: this reconciles an underlying strong coupling with the WIMP miracle. For instance, for $D$=6, considering that in the relevant non-relativistic limit $E \sim m_{\rm DM}$,

$$\Omega_{\rm DM} h^2 \approx 0.1 \left( \frac{4\pi}{g_*} \right)^4 \left( \frac{5\,{\rm GeV}}{m_{\rm DM}} \right)^2 \left( \frac{M}{3\,{\rm TeV}} \right)^4 , \tag{3}$$

showing that even an extremely strongly coupled system $g_* \approx 4\pi$, can reproduce the observed relic abundance, as long as the mediator scale $M$ is in the multi-TeV region.

At high-energy $E \lesssim M$, DM interacts strongly with itself and with the SM, Eq. (2). This is in fact very appealing for the LHC which, operating at high-energy, has direct access to the strongly coupled regime. Moreover, in this regime, the signal from the strongly coupled sector is expected to be strong, and dominate over the LHC irreducible backgrounds (such as $jZ \to j\nu\nu$). For this reason, because large effects can be obtained even for $E \lesssim M$, DM from a strongly coupled sector provides one of the few examples where the use of a DM Effective Field Theory (EFT) is well motivated even to parametrize LHC DM searches - a topic that has received enormous attention in recent years (see Refs. [2–4] and the literature that followed).

In this note we will use symmetry arguments to discuss all structured scenarios where DM is strongly coupled, but fulfills the WIMP miracle. After identifying the relevant symmetries, we use simple power counting rules to build the EFT describing the physics of these scenarios at collider energies, both in the case where DM is a scalar or a fermion. We will see that, in some cases, the EFT for strongly coupled DM differs substantially from the original DM EFT of Refs. [2–4].

[1]Notice that, for simplicity, we limit the present discussion to s-wave annihilation. Annihilation in p-wave would imply in Eq. (1) the presence of the suppression factor $v_{\rm rel}^2$ due to the relative velocity of the two annihilating particles, roughly $v_{\rm rel} \sim 1/3$ at freeze-out temperature.

## 2 Symmetries

So, what symmetries are compatible with irrelevant operators only? For scalars a well-known example is the shift symmetry associated with Nambu–Goldstone bosons (NGBs) from strong dynamics, like QCD pions. In this case the leading interactions appear at $D{=}6$ or $D{=}8$. For Dirac fermions, on the other hand, chiral symmetry and the absence of gauge interactions are enough to guarantee $D \geq 6$. Alternatively, for Majorana fermions (in analogy with NGBs), non-linearly realized supersymmetry (SUSY) ensures that $D \geq 8$. Indeed the leading interactions of Goldstini from spontaneously broken SUSY only exhibit higher-derivative interactions in the limit where all other SUSY particles are heavy [5].

We will discuss these examples in detail below, but first we want to answer the question of whether, beyond these examples, we can find an infinite set of symmetries such that the low-energy amplitude is suppressed by higher and higher powers of energy, i.e. where $D \geq 10$ constitute the only interactions allowed in the limit of exact symmetry. As a matter of fact the answer is negative. Fundamental principles based on analyticity, unitarity and crossing symmetry of the $2 \to 2$ amplitude provide strict positivity constraints for some of the coefficients of $D{=}8$ operators [6]. This implies that generally there is no limit in which a symmetry that protects operators with four fields and $D \geq 10$, forbidding $D \leq 8$, can be considered exact. So the complete set of scenarios with a naturally light strongly coupled DM, that however appears weakly coupled at small $E$ (and therefore fulfills the WIMP miracle) is given by the above examples[2] and is captured by operators of $D \leq 8$.

## 3 Scalar Dark Matter

Naturally light scalars originate as pseudo-NGBs of the spontaneously symmetry breaking (SSB) pattern $G/H$. If the sector responsible for SSB is strong, NGB interactions become strong at high-$E$. These scenarios are particularly interesting in association with the hierarchy problem [9–15], but also independently from it [16, 17]. Qualitatively different cases of interest can be identified, depending on the particular group structure being considered and the interplay with Higgs physics. First, a light scalar DM can be associated with an abelian $U(1) \to \mathcal{Z}_2$ breaking pattern, while a light composite Higgs originates from e.g. $G/H = SO(5)/SO(4)$ [18]. Alternatively, the DM originates from a non-abelian, e.g. $SU(2) \to U(1)$ or larger, symmetry breaking patterns [9, 14–16]. Finally, both the Higgs and DM can arise together from a non-factorizable group $G$, such as $SO(6)/SO(5)$ [10, 11, 13, 19]. The very power of EFTs is that, at low-$E$, large groups of theories fall in the same universality classes: in our case the generic EFTs that we will now build to describe the above-mentioned scenarios can be matched to any model with approximate symmetries.

In all these cases, the NGB interactions are described by the CCWZ construction [20]: the light degrees of freedom $\phi^a$ are contained in the coset representative $U = \exp(i\phi^a t^a/f) \in G/H$ and appear in the Lagrangian only[3] through the building blocks $d_\mu^a$ and $\varepsilon_\mu^A$ in $U^{-1}\partial_\mu U = i d_\mu^a t^a + i\varepsilon_\mu^A T^A$, where $t^a (T^A)$ are the broken (unbroken) generators in $G$, $f$ is the analog of the pion decay constant and is related to the mass and couplings of resonances from the (strong) sector that induces SSB through the naive dimensional analysis

[2] Ref. [7] proposes a somewhat different realization of the same principle, where symmetries imply suppression of the $2 \to 2$ amplitude in favor of the $3 \to 2$, which decouples fast as the DM density dilutes. Alternatively, selection rules in the UV could imply p- or d-wave suppressions in the non-relativistic limit, also satisfying our high-energy/strong-coupling, low-energy/weak-coupling dichotomy (yet the implied cross section suppression is mild $\sim 0.2 \div 0.1$ [8]).

[3] We will assume that anomalies and the Wess-Zumino-Witten term, that might lead to DM decay in similarity to $\pi \to \gamma\gamma$ in QCD, vanish.

| $G/H$ | $\phi$ | $d^a_\mu$ | $\varepsilon^a_\mu$ |
|---|---|---|---|
| $\frac{U(1)}{\mathcal{Z}_2}$ | $\phi \in \mathbb{R}$ | $\frac{\partial_\mu \phi}{f}$ | $0$ |
| $\frac{SU(2)}{U(1)}$ | $\phi \in \mathbb{C}$ | $(1+\frac{\|\phi\|^2}{f^2}+\ldots)\frac{\partial_\mu\phi}{f}$ | $\frac{\phi^\dagger \overleftrightarrow{\partial}_\mu \phi}{f^2}+\ldots$ |
| $\frac{SO(6)}{SO(5)}$ | $H^i, \phi \in \mathbb{R}$ | $\left(1+\frac{\|\phi\|^2}{f^2}+\frac{\|H\|^2}{f^2}+\ldots\right)\frac{\partial_\mu\phi}{f}$ | $\frac{H^\dagger \overleftrightarrow{\partial}_\mu H}{f^2}+\ldots$ |

Table 1: Building blocks for the effective Lagrangian with different SSB patterns. Dots denote higher order terms in $1/f$.

estimate $f = M/g_*$ [21]. Table 1 shows some specific examples.

Under a transformation $g \in G$, $U \to gUh(\phi, g)^{-1}$, where $h(\phi, g) \in H$. Then $d_\mu \equiv d^a_\mu t^a$ and $\varepsilon \equiv \varepsilon^A_\mu T^A$ transform under $G$ respectively in the fundamental representation of $H$ and shift as a connection, so that $D^\varepsilon_\mu \equiv \partial_\mu + i\varepsilon_\mu$ is the covariant derivative. With these ingredients, the low energy Lagrangian describing the canonically normalized light scalars only, is simply $\mathcal{L}^{eff} = M^2 f^2 \mathcal{L}\left(d^a_\mu/fM, D^\varepsilon_\mu/M\right)$, with the additional requirement of invariance under the unbroken group $H$: this automatically guarantees also $G$ invariance.

Clearly DM cannot be an exact massless NGB: the global symmetry must be broken explicitly. We keep track of this breaking by weighting interactions that violate the CCWZ construction with $m^2_\phi/M^2$; an assumption that reflects to good extent the expectations in explicit models (see for instance [10]). We further assume the most favorable case in which, to the extent possible, the SM itself is part of the strong dynamics, as discussed in Ref. [22],[4] so that DM-SM interactions do not introduce further symmetry breaking effects (we discuss below cases where only some species take part in the new dynamics). This implies in particular that we assume the new dynamics respects the SM (approximate) symmetries: custodial symmetry, CP, flavor symmetry (broken only by the SM Yukawas [26]) and baryon and lepton numbers. Finally we assume the new dynamics can be faithfully described by a single new scale $M$ and coupling $g_*$ [23]. Compatibly with these assumptions, the most general Lagrangian at the leading $D = 6$ order in the $1/M$ expansion is,

$$
{}_6\mathcal{L}^{DM_\phi}_{\rm eff} = c^V_\psi \frac{g_*^2}{M^2}\phi^\dagger \overleftrightarrow{\partial}_\mu \phi\, \psi^\dagger \bar\sigma^\mu \psi + c^{dip}_B \frac{g_*}{M^2}\partial_\mu \phi^\dagger \partial_\nu \phi\, B^{\mu\nu}
+ c^S_H \frac{g_*^2}{M^2}|\partial_\mu \phi|^2 |H|^2 + c^{\not s}_H \frac{g_*^2 m^2_{\phi,H}}{M^2}|\phi|^2 |H|^2 + c^{\not s}_\psi \frac{g_*^2 y_\psi}{M^2}|\phi|^2 \psi\psi H\,, \quad (4)
$$

where each operator is weighed by the maximum coefficient that we can expect following the power-counting rules associated with the above mentioned-symmetries. The scaling in powers of the coupling $g_*$ can be unambiguously determined from a bottom-up perspective by restoring $\hbar \neq 1$ in the Lagrangian [21, 23, 27]: the coefficient $c_i$ of an operator $\mathcal{O}_i$ with $n$ fields scales as $c_i \sim (\text{coupling})^{n-2}$.

[4] This implies that the Higgs is itself a PNGB [18,23], SM fermions are partially composite [18,24], and the transverse polarizations of gauge bosons have strong multipolar interactions [22] – constraints on these possibilities, independent of the new sector couplings to DM, will be studied in [25].

Similarly, at $D$=8, focussing on operators that contribute to $2 \to 2$ scattering,

$$
{}_8\mathcal{L}_{\rm eff}^{DM} = C_V^{\not{S}} \frac{g_*^2 m_\phi^2}{M^4} |\phi|^2 V_{\mu\nu}^a V^{a\,\mu\nu} + C_\psi^S \frac{g_*^2 y_\psi}{M^4} |\partial^\mu \phi|^2 \psi\psi H
+ C_V^S \frac{g_*^2}{M^4} |\partial^\mu \phi|^2 V_{\nu\rho}^a V^{a\,\rho\nu} + C_H^S \frac{g_*^2}{M^4} |\partial^\mu \phi|^2 |D^\nu H|^2
+ C_V^T \frac{g_*^2}{M^4} \partial^\mu \phi^\dagger \partial^\nu \phi V_{\mu\rho}^a V_\nu^{a\,\rho} + C_H^T \frac{g_*^2}{M^4} \partial^\mu \phi^\dagger \partial^\nu \phi D_{\{\mu} H^\dagger D_{\nu\}} H
+ C_\psi^T \frac{g_*^2}{M^4} \partial^\mu \phi^\dagger \partial^\nu \phi \psi^\dagger \bar{\sigma}_\mu D_\nu \psi \,, \tag{5}
$$

with $V_{\mu\nu}^a = B_{\mu\nu}, W_{\mu\nu}^a, G_{\mu\nu}^a$ for $U(1)_Y \times SU(2)_L \times SU(3)_C$ gauge bosons, and $\psi$, $H$ the SM fermions and Higgs. We use a notation based on left-handed Weyl fermions, which carry additional internal indices to differentiate left-handed $\psi$ and right-handed $(\psi^c)^\dagger$ components of Dirac fermions [8]; the Wilson coefficients $c, C$, associated to the $D = 6, 8$ Lagrangians respectively, carry these indices, and are expected to be $O(1)$, unless otherwise stated, see table below.

Of course there are more operators that contribute to $2 \to 2$ scattering, but these can either be eliminated through partial integration, field redefinitions (that eliminate operators proportional to the equations of motion), Bianchi or Fierz identities[5], or they violate some of the linearly realized symmetries that we assume (CP, custodial). For instance, operators antisymmetric in the Higgs field, such as

$$
c_H^{\not{cust}} \frac{g_*^2}{M^2} \phi^\dagger \overleftrightarrow{\partial}_\mu \phi \, H^\dagger \overleftrightarrow{D}^\mu H \tag{6}
$$

transform as $(\mathbf{1},\mathbf{3})$ under custodial symmetry $SU(2)_L \times SU(2)_R$: their coefficient is expected to be generated first at loop level by custodial breaking dynamics, involving for instance $g'$, which satisfies the required transformation rules $c_H^{\not{cust}} \sim g'^2/16\pi^2$. On the other hand at $D$=8,

$$
\partial^\mu \phi^\dagger \overleftrightarrow{\partial}_\nu \partial_\mu \phi H^\dagger \overleftrightarrow{D}^\nu H \,, \qquad \partial^\mu \phi^\dagger \overleftrightarrow{\partial}_\nu \partial_\mu \phi \psi^\dagger \bar{\sigma}^\nu \psi \,, \tag{7}
$$

share the same symmetries (among the linearly and non-linearly realized ones that we have presented[6]) as operators in ${}_6\mathcal{L}_{\rm eff}^{DM_\phi}$ and contribute to the same observables; for this reason their contribution is expected to be always suppressed by $\sim E^2/M^2 \ll 1$ in the amplitude and we neglect them ( a similar logic was followed in Ref. [29] to argue that the Peskin-Takeuchi [30] $U$-parameter can be neglected, since it shares the same symmetries as the $T$ parameter, but is higher-dimension).

Similarly, $m_\phi^2 |\phi|^2 |H|^4$ and $\partial_\mu \phi^\dagger \partial^\mu \phi |H|^4$ give a subleading (by a factor $g_*^2 v^2/M^2 \lesssim 1$) contribution w.r.t. $c_H^S$ and $c_H^{\not{S}}$, in processes with 2 longitudinal vectors or Higgses and can only be distinguished in processes with three or more external longitudinal vector bosons/Higgses. Finally, operators of the form $|\phi|^2 \times {}_6\mathcal{L}_{\rm eff}^{SM}$, where ${}_6\mathcal{L}_{\rm eff}^{SM}$ is the $D$=6 SM Lagrangian (see Ref. [31]) but also includes total derivatives, are generally further suppressed by $m_\phi^2/M^2$ and count as $D$=10 effects in our perspective.

The important novel aspect that is emphasized by our analysis and summarized in the Lagrangians Eqs. (4,5) and table 1, is the following. Both the $D$=6 and $D$=8 Lagrangians can

[5] We eliminate structures involving $\sigma^{\mu\nu}$ in favor of structures that can be generated by tree-level exchange of scalars or vectors.

[6] Technically the set of infinite symmetries of the free Lagrangian $\hat{\phi}(p) \to e^{i\theta(p)}\hat{\phi}(p)$, $\theta(-p) = -\theta(p)$ in momentum space, is broken by the operators of Eq. (7) and those in Eqs. (4,5) to different subgroups, so that a Lagrangian with only the interactions of Eq. (7) is technically natural per se [28]; yet, it is incompatible with the positivity constraints mentioned in the introduction.

| | $c_\psi^V$ | $c_B^{dip}$ | $c_H^S$ | $c_H^{\not{\partial}}$ | $c_\psi^{\not{\partial}}$ | $C_V^{S,T}$ | $C_\psi^T$ |
|---|---|---|---|---|---|---|---|
| $\psi_{elem}$ | × | | | | × | | × |
| $V_{elem}$ | | × | | | | × | |
| $U(1)/\mathcal{Z}_2$ | × | × | × | × | × | | |
| $SU(2)/U(1)$ | | | × | × | × | | |
| $SO(6)/SO(5)$ | × | × | | | | | |

Table 2: × denotes suppression of a given EFT coefficient, according to specific properties of the microscopic dynamics: $\psi_{elem}$ denotes the limit where SM fermions are not composite, $V_{elem}$ denotes instead the familiar case where the transverse polarizations of vectors are elementary (as opposed to strong multipolar interactions [22]).

be important, as symmetries can suppress the expected leading interactions in favor of higher order ones. Indeed, as table 1 shows, the structures $c_\psi^V$ vanishes for antisymmetry if DM has a single real degree of freedom (such as for the $U(1)/\mathcal{Z}_2$ and $SO(6)/SO(5)$ cosets), so that in this case the leading DM-fermion interaction is given by the $D$=8 operator $C_\psi^T$. On the other hand the structures $c_\psi^{\not{\partial}}$ and $c_H^S$ are unsuppressed only when the generators associated with $\phi$ and $H$ do not commute (such as in the $SO(6)/SO(5)$ model [10, 19]), but will be further suppressed by $\sim m^2_{\phi,H}/M^2$ in other cases. In those cases the leading DM-Higgs interactions are the $D$=8 $C_H^S$ and $C_H^T$. Finally, an important source of suppression is represented by the degree of compositeness of the SM particles - either fermions or (transverse) gauge bosons. The most favorable situation is when the SM particles are fully composite since in this case they feature an unsuppressed $g^*$ coupling to the strong sector. On the contrary, if SM fermions and gauge bosons are elementary degrees of freedom, we expect a suppression in the corresponding couplings, as shown in the first two rows of table 2.[7] In models where the DM dominantly couples to gluons only, the leading effects at high-energy, not suppressed by any small parameters, are the $D$=8 $C_V^S$ and $C_V^T$. We summarize in table 2 these and other such situations, where some of the above operators are suppressed by additional small parameters (such as symmetry breaking effects), and become therefore less interesting from the point of view of collider searches.

In Fig. 1 we compare the LHC reach (blue region) in the $(g_*, M)$-plane with relic density (RD) expectations (green band) for $D = 6$ (e.g. DM as a PNGB of $SU(2)/U(1)$), showing that visible LHC effects are compatible with a non-vanishing RD. Here the LHC constraints have been derived from the data of Ref. [35], imposing an additional cut in the centre-of-mass energy $\hat{s} < M^2$. This cut, and the representation in the $(g_*, M)$-plane, help us establishing consistency of the EFT assumption [27, 36, 37]. Indeed, as $M$ is lowered within the LHC kinematic region, the constraints rapidly deteriorate, since less and less data remains available: this signals the fact that, in that region, our EFT assumptions are not verified. See a companion paper [8] for more details.

LHC constraints for the examples discussed above, where $D$=8 represent the leading effect at high-$E$, are also shown in Fig. 1 with a dashed (red) curve. Notice that here, while the $E$-growing cross sections implied by our symmetry structure clearly dominate at LHC energies $M \gtrsim E \gg m_{\rm DM}$, they might be comparable to symmetry breaking $m_{\rm DM}$-suppressed interaction at low-$E$, relevant at freeze-out. In other words, the complementarity between different DM experiments is partially lost in this setup – we discuss this issue further in [8].

[7] A more realistic situation is when only the right-handed top quark is fully composite [32,33]: see Ref. [34] for a discussion of DM in this case.

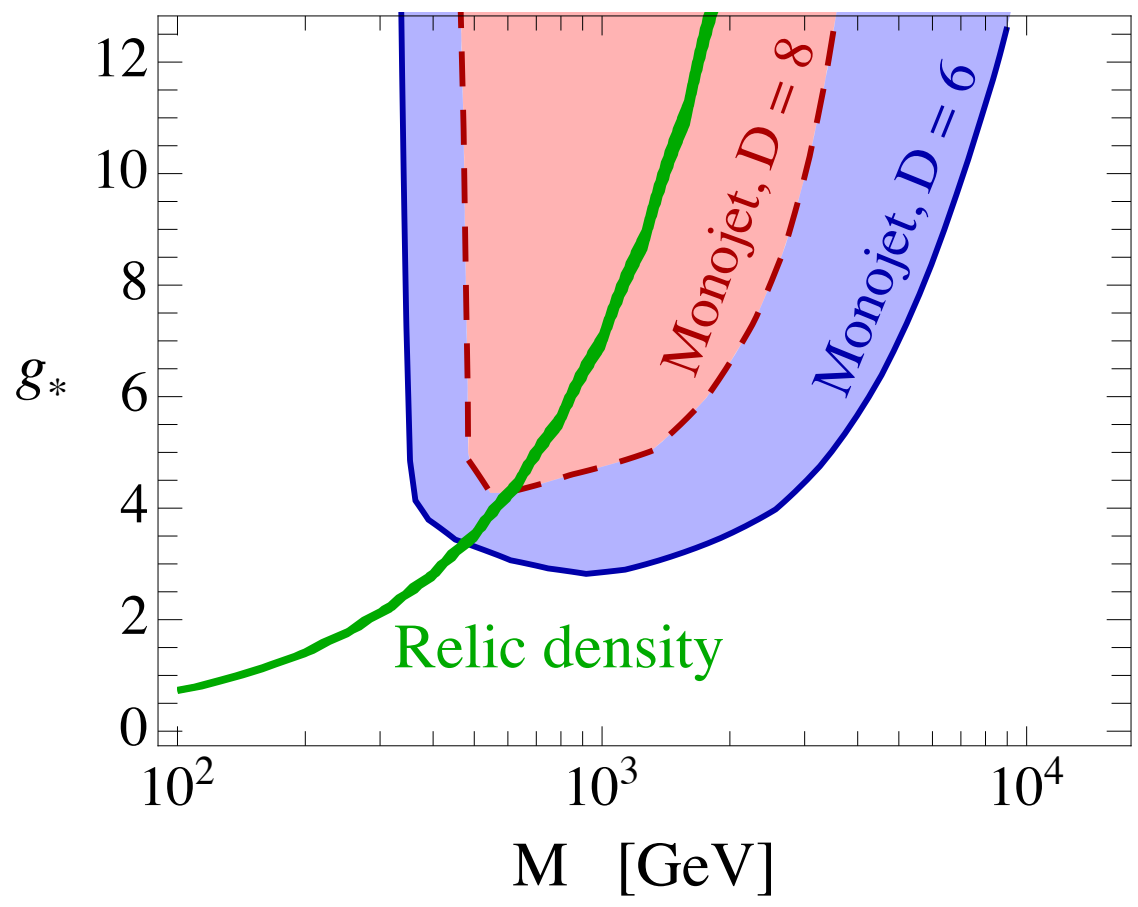

Figure 1: Constraints on scalar DM with $m_{\rm DM} = 5$ GeV. Blue region: excluded by consistent LHC constraints on $D$=6 operator $c_\psi^V$ in Eq. (4) (e.g. pseudo-NGB DM from a non-abelian SSB pattern), and comparison with the parameters where the RD is correctly reproduced with the same D=6 operator (solid green). Red region: LHC constraints on $D$=8, $C_\psi^T$ in Eq. (5) (e.g. one scalar DM from an abelian SSB).

## 4 Fermionic Dark Matter

As mentioned above, if DM is a strongly interacting fermion $\chi$ there are two structurally robust situations in which its mass and low-energy interactions might appear small: chiral symmetry for Dirac fermions or non-linearly realized SUSY for Majorana fermions. The first case is familiar: interactions involving the product $\chi^\dagger_{\dot\alpha}\bar\sigma^{\dot\alpha\alpha}_\mu\chi_\alpha$ preserve chiral symmetry, while $\chi_\alpha\chi^\alpha$ break it and are expected to be weighed by $m_{\rm DM}/M$.

In the second case, DM fermions are Goldstini of non-linearly realized SUSY. There are different motivations to discuss this scenario. First of all, a supersymmetric version of the equivalence theorem [38] implies that in the high-energy limit $E \gg m_{3/2}$, the gravitino behaves effectively like a Goldstino (in this case, however, the relation $m_{3/2} \sim F/M_{\rm Pl}$ implies – for a SUSY breaking sector at $\sqrt{F} \sim$ TeV, necessary to have sizable LHC effects – a very light gravitino). Goldstini are even more interesting in scenarios where $\mathcal{N} = 1$ SUSY is spontaneously broken in $n > 1$ nearly sequestered sectors [39]: in this case $n-1$ approximate Goldstini appear in the light spectrum and are good DM candidates (their mass being independent from the strength of their interactions). More generally, in an EFT perspective, we can consider the case of approximate $\mathcal{N} \geq 1$ SUSY that, when spontaneously broken, includes light Goldstini in the spectrum [40], and these are good DM candidates.

We work in the simplified limit where all SUSY partners are heavy $m_{susy} \approx \sqrt{F}$ so that the physics of Goldstini at $E \ll \sqrt{F} \equiv M$ can be described in a formalism that parallels the CCWZ construction [41, 42], adapted to the breaking of spacetime symmetries [43–45]. The coset representative can be written as

$$U = e^{iPx} e^{i\frac{\chi}{2}Q} e^{i\frac{\chi^\dagger}{2}\bar{Q}}\,, \tag{8}$$

where $Q$ and $\bar{Q}$ are the SUSY generators, $\chi$ the Goldstino, and the presence of momenta $P$ is a peculiarity of spontaneously broken space-time symmetries (it can be somehow thought as due to the fact that translations themselves are realized through coordinates shifts, in a way

that mimics non-linear realizations [45]). The Maurer-Cartan form is now

$$U^{-1}\partial_\mu U = i\left(\delta^a_\mu + \frac{i}{2}\partial_\mu\chi\sigma^a\chi^\dagger\right)P_a + \frac{i}{2}\partial_\mu\chi Q + \frac{i}{2}\partial_\mu\chi^\dagger\bar{Q}\,.$$

Here the important building block of the low-energy Lagrangian is $E^a_\mu \equiv \delta^a_\mu + \frac{i}{2}\partial_\mu\chi\sigma^a\chi^\dagger$, which transforms as a vierbein and plays the analogous rôle as $\varepsilon_\mu$ for NGBs, rendering a Poincarré-invariant action, written in terms of these building blocks, into one invariant under (non-linear) SUSY. In particular, $\int dx^4 F^2 \det E^a_\mu = i\chi^\dagger\bar{\sigma}^\mu\partial_\mu\chi + \cdots$, contains the kinetic term for the canonically normalized Goldstino [46], while interactions with light matter can be described through the vierbein $E^a_\mu(\chi)$ and metric $g_{\mu\nu}(\chi) \equiv E^a_\mu(\chi)E^a_\nu(\chi)$. For our purpose, the important result is that interactions with light fermions $\psi$, scalars $\phi$ or gauge field strengths $F_{\mu\nu}$ are captured by the following $D$=8 operators:

$$\begin{aligned} &\frac{1}{F^2}\chi^\dagger\bar{\sigma}^\mu\partial_\nu\chi\bar{\psi}\bar{\sigma}_\mu\partial^\nu\psi\,, \qquad \frac{1}{F^2}\chi^\dagger\bar{\sigma}^\mu\chi\partial_\nu\bar{\psi}\bar{\sigma}^\mu\partial^\nu\psi\,, \\ &\frac{i}{F^2}\chi^\dagger\bar{\sigma}^\mu\partial^\rho\chi\, F_{\mu\nu}F_\rho^{\ \nu}\,, \qquad \frac{i}{F^2}\chi^\dagger\bar{\sigma}^{\{\nu}\partial_{\mu\}}\chi\,\partial_\nu H^\dagger\partial^\mu H\,. \end{aligned} \tag{9}$$

If $\psi$, $F$ or $H$ are part of the strong sector, the coefficients of some of these operators are related by supersymmetry to their kinetic terms and depend on the scale $F$ only; in what follows we will leave them as free parameters, entertaining the possibility that SM states by partially composite, in which case the operators Eq. (9) will be proportional the the composite-elementary sectors mixing.

An explicit Goldstino mass can only be associated with explicit SUSY breaking (or departures from exact sequestering in [39]), which will generate operators different from Eq. (9), suppressed by $m_{\rm DM}/M$. Similarly to the scalar case above, we use this fact and power-counting arguments to write the most general effective Lagrangian weighed by the strongest possible interaction that can be achieved in the scenarios under scrutiny, and postpone more restrictions below.

At leading order in the $1/M$ expansion, the effective Lagrangian for fermionic DM reads

$${}_6\mathcal{L}^{DM}_{\rm eff} = c^V_\psi\frac{g_*^2}{M^2}\chi^\dagger\bar{\sigma}^\mu\chi\psi^\dagger\bar{\sigma}_\mu\psi + c^S_H\frac{g^2_{\rm SM}m_\chi}{M^2}\chi\chi H^\dagger H + c^{dip}_B\frac{g_* m_\chi}{M^2}\chi\sigma^{\mu\nu}\chi B_{\mu\nu}\,, \tag{10}$$

where the coefficient of $c^S_H$ reflects the fact that it does not respect the Higgs NGB symmetry (recall that in order for the Higgs to take part in the strong dynamics and be light, it is expected to arise as a PNGB [23]) and can only arise via effects involving SM symmetry breaking couplings, that we denote generically as $g_{\rm SM}$. At $D$=8 we find,

$$\begin{aligned} {}_8\mathcal{L}^{DM}_{\rm eff} = C^{\not{s}}_\psi\frac{m_\chi y_\psi g_*^2}{M^4}\chi\chi\psi\psi H + C^{\not{s}\prime}_\psi\frac{m_\chi y_\psi g_*^2}{M^4}\chi\psi\psi\chi H \\ + C^{\not{s}}_V\frac{g_*^2 m_\chi}{M^4}\chi\chi V^a_{\mu\nu}V^{a\mu\nu} + C_V\frac{g_*^2}{M^4}\chi^\dagger\bar{\sigma}^\mu\partial^\nu\chi V^a_{\mu\rho}V^{a\rho}_\nu \\ + C_\psi\frac{g_*^2}{M^4}\chi^\dagger\bar{\sigma}^\mu\partial^\nu\chi\psi^\dagger\bar{\sigma}_\mu D_\nu\psi + C'_\psi\frac{g_*^2}{M^4}\chi^\dagger\bar{\sigma}^\mu\chi D^\nu\psi^\dagger\bar{\sigma}_\mu D_\nu\psi \\ + C_H\frac{g_*^2}{M^4}\chi^\dagger\bar{\sigma}^\mu\partial^\nu\chi D_\mu H^\dagger D_\nu H\,. \end{aligned} \tag{11}$$

For generic Wilson coefficients, Eqs. (10,11) represent the most general $D$=6,8 contributions to $2\to 2$ on-shell scattering at $D\leq 8$ (for $D$=6 see also [47]). Other structures either violate underlying symmetries or can be eliminated as described in the scalar case above. In particular

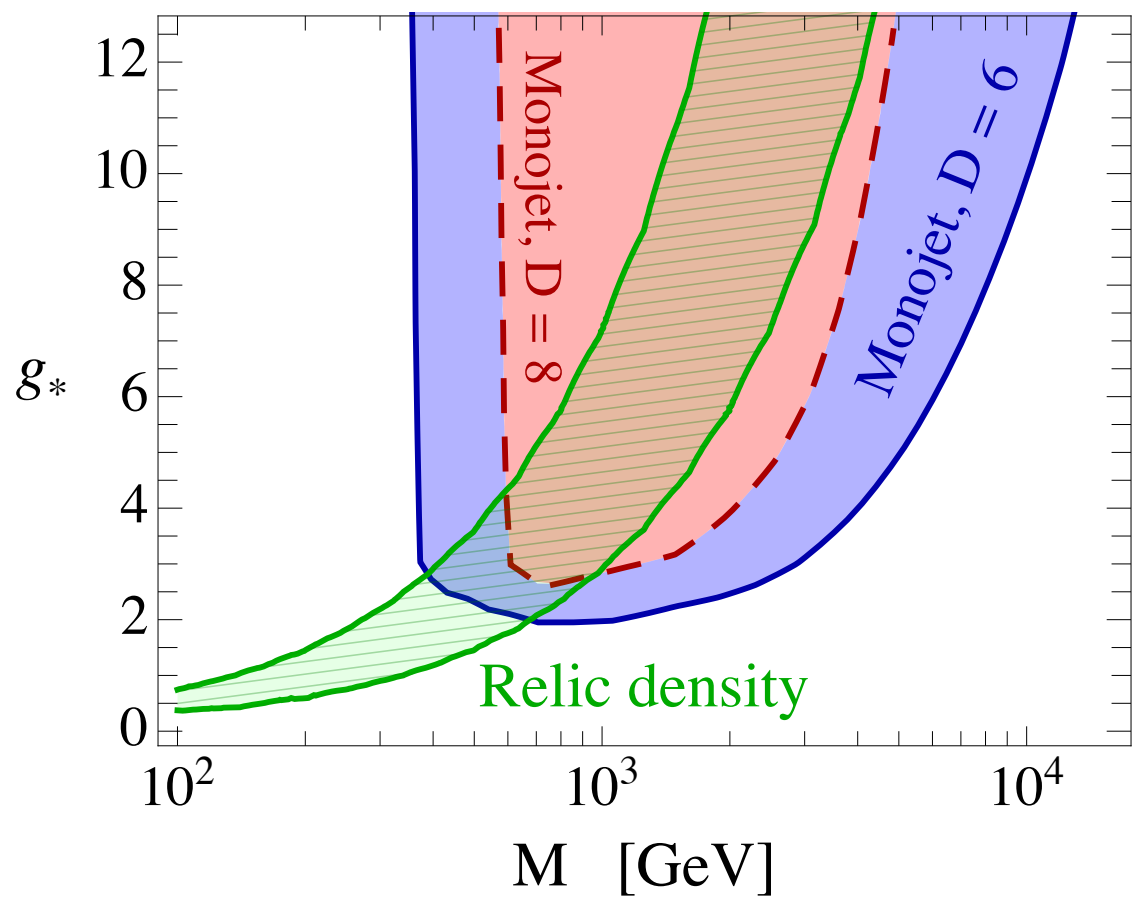

Figure 2: Same as in Fig. 1, but for fermionic Dirac DM. The region shaded in green corresponds to the observed relic abundance (fermionic DM comes with incertitude about the chiral structure of the $D = 6$ effective operator considered [8], reflected by the width of the band, contrary to the single line of the scalar case).

it can be shown that only three hermitian operators of the form $D^2\psi^4$ exist at $D$=8 and one, corresponding to the imaginary part of $C_\psi$ in Eq. (11), violates CP and we neglect it. Similarly operators antisymmetric in $H \longleftrightarrow H^\dagger$, like $\chi^\dagger\bar{\sigma}^\mu\chi H^\dagger \overleftrightarrow{D}_\mu H$ that plays a rôle in mono-Higgs searches [48], violate custodial symmetry and we neglect them. Moreover, operators of the form $|H|^2 \times {}_6\mathcal{L}^{eff}$ also appear at $D = 8$, but, similarly to the scalar DM case, they are are expected to be small (if the Higgs is also a PNGB) and moreover they only affect processes with additional $h$.

So, for composite Dirac fermions, only the $D = 6$ $c_\psi^V$ is important (also, for light DM $c_B^{dip}$ and $c_H^S$ are constrained by constraints from $Z$ and $h$ decays) in Fig. 2 we show that the LHC is here providing the most important piece of information, accessing the region in parameter space that reproduces the observed RD.[8] Notice that the latter depends on the specific chiral structure of the $D = 6$ effective operator considered. To be more specific, interactions involving a vector (axial-vector) coupling in the DM current are characterized by an unsuppressed s-wave (p-wave suppressed) annihilation cross section, and the observed amount of RD corresponds to the lower (upper) curve in the green band shown in Fig. 2.

Nevertheless, if $\chi$ is a Goldstino, then the $D = 6$ Lagrangian vanishes in the limit of exact SUSY, and the first strong interactions appear at $D = 8$. In this case only $C_V$ and $C_\psi^{(\prime)}$ are important for mono-jet analyses. This is an example (similar to the $U(1)/\mathcal{Z}_2$ PNGB) where approximate symmetries, that were invoked to hide strong coupling at small energy, go as far as suppressing the first order $1/M^2$ terms but allow the $1/M^4$ ones. Even in this case the LHC contains important information (dashed line of Fig. 2).

## 5 Outlook

In Summary, we have discussed natural situations in which light DM originates from a strongly-coupled sector but its interactions are small at low-energies because of approximate symmetries, that forbid relevant interactions and allow only irrelevant (higher-derivative) ones.

[8] In addition to our $E$-scaling, renormalization group effect can play a rôle in the precise comparison between LHC and RD probes (see e.g. [49]).

Prime principles dictate that such symmetries are consistent only with $D$=6 and $D$=8 operators for 2 → 2 scattering. In this article we have identified generic effective Lagrangians at these orders and introduced a power-counting that captures the most well-motivated scenarios that can imply large effects in irrelevant interactions: scalar DM as a PNGB, or fermion DM as a strongly coupled fermion or Goldstino.

These provide a class of models in which the LHC high-$E$ reach plays an important rôle with respect to other types of experiments (such as RD indications and direct detection) and contains genuinely complementary information. Moreover, in these scenarios the DM EFT is not only consistent with LHC analysis (due to the underlying strong coupling, as shown in Figs. 1 and 2), but also necessary, as the underlying dynamics is uncalculable. Our characterization provides a well-motivated context to model missing transverse-energy distributions at the LHC, in mono-jet, mono-W,Z,$\gamma$ or mono-Higgs searches, with a handful of relevant parameters and yet a clear and consistent microscopic perspective. To the question of what we have learned from LHC DM searches, these models provide one answer.

## Acknowledgments

We acknowledge important conversations with Brando Bellazzini (whom we also thank for comments on the manuscript), Roberto Contino, Rakhi Mahbubani, Matthew McCullough, Alexander Monin, Alex Pomarol, Davide Racco, Riccardo Rattazzi, Andrea Wulzer.

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
