# Peer review of "Strongly Interacting Light Dark Matter"

_SciPost Physics, doi:SciPost Phys. 3, 017 (2017)_

## Round 2 · Referee Report · Tim Tait · 2017-4-21

Strengths

1. This paper looks at a well-thought out parameterization of a class of a few classes of strongly interacting models for the dark matter.
2. The results are new, and interesting.

Weaknesses

1. There are only very minor presentation weaknesses outlined below.

Report

In general, this is a solid contribution to the literature of composite dark matter, and it should be published after small revision described below.

Requested changes

1. A few comments in the paper are strange. For example, toward the end of the first section the comment is made: "(see [4] and the literature that followed)." is odd because the papers in References [3] and [5] all pre-date those in [4], so it is more correct stay that [4] followed [3]. This statement must be made causally correct.
2. Also at the end of the Motivation section, the authors write "Surprisingly, we will see that the EFT for strongly coupled DM differs substantially from the original DM EFT of Refs. [3-5]". This statement is not really correct. The pNGB and Goldstino-like DM sections do differ because they assume very different symmetries of the underlying theory -- thus, it is true that they differ, but it is hardly surprising. In the case of the more general fermionic DM, the operators have a large correspondence with those in cited references (though the power counting is handled differently).
3. The discussion of the extraction of the LHC limits should not be buried in a footnote and needs more detail.

  • validity: high
  • significance: good
  • originality: good
  • clarity: ok
  • formatting: good
  • grammar: excellent

Author:  Francesco Riva  on 2017-04-21  [id 119]

(in reply to Report 1 by Tim Tait on 2017-04-21)

We thank the referee for the constructive comments.

  1. We have reordered the references and referred to them as a whole, rather than singling out Bai et al. (which makes sense, given they were written very close to eachother)

  2. Correct, we have rephrased that sentence as “We will see that, in some cases, the EFT for strongly coupled DM differs substantially from the original DM EFT of …”

  3. Yes, it is certainly clearer in the text. We have removed the footnote and expanded it in the text: “Here the LHC constraints have been derived from the data of Ref. [35], imposing an additional cut in the centre-of-mass energy $s < M^2$ . This cut, and the representation in the $(g_∗ , M )$- plane, help us establishing consistency of the EFT assumption [27, 36, 37]. Indeed, as M is lowered within the LHC kinematic region, the constraints rapidly deteriorate, since less and less data remains available: this signals the fact that, in that region, our EFT assumptions are not verified. See a companion paper [8] for more details. ”

Attachment:

StrongDMShort.pdf

---

## Round 2 · Referee Report · Brando Bellazzini · 2017-4-24

Strengths

1. This neat paper identifies the generic features of a broad class of Dark Matter models exploiting the power of EFT.
2. Selection rules, symmetries, scaling and power counting are identified and clearly explained, making the results very general and nicely organized.
3. The use of an approximate supersymmetry for the strongly coupled fermionic DM setup is new and extremely original.

Weaknesses

I find almost no weakness in this paper. I have only a few comments explained in the report section.

Report

This paper is a nice, short, and original paper about a broad class of models where the DM particle emerges from a strong sector. The focus is on the impact of different selection rules and approximate symmetries on the interpretations of the bounds at the LHC, using the framework of EFT.

I find the paper relevant and original, particularly so for the use of an approximate and spontaneously broken SUSY where the DM particle is identified with a pseudo-Goldstino. While its leading interactions are robustly determined by a SUSY-breaking power counting, the resulting peculiar energy-scaling induced by dim-8 operators weakens the reach of the searches at the LHC.

I have only a couple of comments and suggestions for the authors.

1. In Eq.( 11) the authors classify the leading pseudo-Goldstino interactions. I find the notation there somewhat misleading.
The explicit SUSY-breaking terms $C^{\slashed{s}}$ carry factors of the Dark matter mass and, possibly, of other spurions that break other symmetries (e.g. the Yukawa for chiral sym.).
In contrast, the other $C_{V,H,\ldots}$ terms carry only strong coupling factors as if they were breaking no symmetry and any value was allowed.
However, for generic values of these $C_{VH,\ldots}$, I find that they would actually break SUSY too. This is the case whenever $C_{H,\psi,\ldots}\neq 1$ and $C_{V}\neq -1/2$. And the fact that those $C$'s could be set to 1, to -1/2, or slightly departure from it, depends on the extra assumption whether the particles $\psi$, $V$ or $H$ are part of the strong sector (i.e. mostly composite), or are instead elementary. In particular, in most of the composite Higgs models the SM gauge bosons are taken elementary and I'd expect therefore $C_V$ to be treated as the $C^{\slashed{s}}$, e.g. factoring out the $g_{SM}^2$ suppression factor. The same remark is probably important for the light quarks since in most of composite Higgs models are as well (mostly) elementary.

I recommend the authors to clarify these points and specify which other particles are assumed to be composite or elementary, and whether these extra assumptions have an impact on the results of Fig.2.

2. The annihilation cross-section that is reported in Eq.1-3 and used to determine the relic abundance is the one for s-wave annihilation (although here there is a slightly conflation of scaling, the highly relativistic one in Eq.2 and the non-relativistic one in Eq.1 and Eq.3).
Is it clear that the Goldstino-DM case corresponds to s-wave rather than p-wave? I think the paper would improve with a comment on what type of annihilation is taking place in general or at least in Fig.2.

Requested changes

I requests the authors to clarify the minor points 1. and 2. that I have outlined in the general report.

  • validity: high
  • significance: good
  • originality: top
  • clarity: good
  • formatting: excellent
  • grammar: good

Author:  Francesco Riva  on 2017-05-09  [id 129]

(in reply to Report 2 by Brando Bellazzini on 2017-04-24)

We thank the referee for the important comments.

1) Indeed SUSY and full compositeness would fix the coefficients in eq.9 and 11. We have added the following text between the two equations, in parallel with the referees comments:

“If $\psi$, $F$ or $H$ are part of the strong sector, the coefficients of some of these operators are related by supersymmetry to their kinetic terms and depend on the scale $F$ only; in what follows we will leave them as free parameters, entertaining the possibility that SM states by partially composite, in which case the operators Eq.(9) will be proportional the the composite-elementary sectors mixing.”

2) We have added comments about p-wave annihilation in footnote 1: "Notice that, for simplicity, we limit the present discussion to s-wave annihilation. Annihilation in p-wave would imply in Eq.(1) the presence of the suppression factor $v_{\rm rel}^2$ due to the relative velocity of the two annihilating particles, roughly $v_{\rm rel} \sim 1/3$ at freeze-out temperature."

and again in the second column of p5:

"Notice that the latter depends on the specific chiral structure of the $D = 6$ effective operator considered. To be more specific, interactions involving a vector (axial-vector) coupling in the DM current are characterized by an unsuppressed s-wave (p-wave suppressed) annihilation cross section, and the observed amount of RD corresponds to the lower (upper) curve in the green band shown in Fig.2."

---

## Editorial Decision

published